# The Association of Dietary Intake, Oral Health, and Blood Pressure in Older Adults: A Cross-Sectional Observational Study

**DOI:** 10.3390/nu14061279

**Published:** 2022-03-17

**Authors:** Pinta Marito, Yoko Hasegawa, Kayoko Tamaki, Ma Therese Sta. Maria, Tasuku Yoshimoto, Hiroshi Kusunoki, Shotaro Tsuji, Yosuke Wada, Takahiro Ono, Takashi Sawada, Hiromitsu Kishimoto, Ken Shinmura

**Affiliations:** 1Division of Comprehensive Prosthodontics, Faculty of Dentistry & Graduate School of Medical and Dental Sciences, Niigata University, Niigata 951-8514, Japan; pintamarito@dent.niigata-u.ac.jp (P.M.); mari18@dent.niigata-u.ac.jp (M.T.S.M.); yoshimoto@dent.niigata-u.ac.jp (T.Y.); ono@dent.niigata-u.ac.jp (T.O.); 2Department of Prosthodontics, Faculty of Dentistry, Universitas Indonesia, Jakarta 10430, Indonesia; 3Department of Dentistry and Oral Surgery, Hyogo College of Medicine, Nishinomiya 663-8501, Japan; kisihiro@hyo-med.ac.jp; 4Department of General Internal Medicine, Hyogo College of Medicine, Nishinomiya 663-8501, Japan; kayoko_tamaki@hotmail.com (K.T.); kusunoki1019@yahoo.co.jp (H.K.); 5Department of Prosthodontics, College of Dentistry, Manila Central University, Caloocan 1400, Philippines; 6Department of Orthopaedic Surgery, Hyogo College of Medicine, Nishinomiya 663-8501, Japan; tj13041305sho@gmail.com; 7Department of Rehabilitation Medicine, Hyogo College of Medicine Sasayama Medical Center, Sasayama 669-2321, Japan; yu-wada@hyo-med.ac.jp; 8Hyogo Dental Association, Kobe 650-0003, Japan; sawada@fc.hda.or.jp

**Keywords:** hypertension, blood pressure, oral health, dietary intake, brief-type self-administered diet history questionnaire

## Abstract

Hypertension is related to impaired mastication that causes malnutrition, declining the general health of older adults. This study assessed the role of dietary intake in the relationship between oral health and blood pressure. Eight hundred ninety-four adults aged ≥65 years who independently lived in rural regions of Japan participated in this study. Hypertension was classified according to the guidelines of the Japanese Society of Hypertension. The oral condition was evaluated by analyzing the remaining teeth, occlusal force, posterior occlusal support, masticatory performance, oral moisture, and oral bacterial level. Dietary intake was assessed using a brief self-administered dietary history questionnaire. Mann-Whitney U, chi-square, Kruskal-Wallis tests, and logistic regression analyses were used to elucidate the factors related to hypertension. Normotensive, hypertensive, and history of hypertension were observed in 30.9%, 23.8%, and 45.3% of the participants, respectively. The factors significantly associated with the hypertension were age, body mass index, posterior occlusal support condition, and sodium-to-potassium ratio related to salt intake and/or vegetable intake. Participants without posterior occlusion significantly had higher risk of hypertension (odds ratio = 1.72). This study suggested that there was an association between oral health and hypertension, while the loss of occlusal support may influence nutritional intake conditions.

## 1. Introduction

The prevalence of hypertension has more than doubled in the last 30 years, making it one of the leading causes of disease and mortality worldwide, with an estimated 1.28 billion people suffering from hypertension in 2019 [1,2]. Hypertension has become one of the leading causes of death in Japan among noncommunicable diseases [3,4]. Alarmingly, the number of people with hypertension in Japan is estimated to be 43 million [1,5], but only 50% received treatment, while 25% had controlled blood pressure (BP) [5]. Furthermore, approximately 70% of older adult population (≥75 years) of Japan has hypertension [6]. According to the Japan Society of Hypertension, individuals who are ≥ 75 years old with a BP ≥ 140/90 mmHg are considered hypertensive [5]. Hypertension is a complex medical condition caused by several factors. Previous studies have identified that periodontal disease [7], occlusal status [8], and tooth loss [8,9,10] are associated with hypertension. However, the role of oral health in hypertension is yet to be clarified. Identifying the risk factors for hypertension, even those with marginal risk, is crucial to devise strategies to prevent the development of hypertension and thus prevent cardiovascular disease. Several studies have reported that hypertension is associated with oral health, including impaired mastication, poor oral hygiene, and oral inflammation [7,9,10,11].

Oral health, which is an indicator of general health, can be affected by a range of diseases and conditions that include dental caries, periodontal disease, and tooth loss [12]. Teeth and oral function constitute the main pathways considered vital in connecting oral to general health. According to the Health 21 plan of Japan, improvement of oral function is the primary target for older adults [13], and the number of teeth has been long established as one of the indicators of the oral health condition [14]. According to the Japan Dental Diseases Survey in 2016, approximately 280,000 and 100,000 patients were estimated to have minor (1–8 missing teeth in one jaw) and major (9–14 missing teeth in one jaw) tooth loss, respectively [15]. Further, adults (≥75 years) lose a minimum of ten teeth per year [16]. A study has reported that tooth loss, particularly in older adults, was associated with malnutrition [17]. Tooth loss invariably leads to the decline of mastication ability, changes in food selection and dietary intake, and changes in nutrient intake, all of which, consequently, have an adverse effect on general health, increasing the risk of systemic diseases, frailty, and mortality [17,18,19,20,21,22].

As oral and general health decline with aging and disease(s) [20,21,23,24,25,26], the number of unchewable food particles increases over time, leading to changes in food selection and eating habits [19,23,24]. According to the 2018 National Health Nutrition Survey in Japan, approximately 25% of people (≥60 years of age) reported a decline in masticatory function, which implies that they were unable to chew a variety of food [13]. Mastication is the first step of the digestive process of breaking food into smaller particles for swallowing that allows more nutrient absorption, which is essential for the maintenance of health, especially in older adults [24,27]. Numerous studies have reported that masticatory function is influenced by several factors, such as the number of remaining teeth [21,28], posterior occlusal contact [21,29,30], occlusal force [21,30], salivary secretion [28,30], and tongue function [28,31]. Tooth loss influences an individual’s food choice and dietary intake, leading to maladaptive behaviors. For example, an individual with tooth loss will prefer eating soft and easy-to-chew foods [32], and will avoid fiber-rich and nutrient-dense foods, such as raw fruits and vegetables, nuts, meats, and grain products, thereby increasing fat, sugar, other carbohydrates, and processed food consumption [32,33].

To measure an individual′s usual food consumption and dietary intake, a food frequency questionnaire can be utilized. Food questionnaires have become one of the main research tools in nutritional epidemiology [34]. In Japan, Sasaki et al. [35] developed a brief self-administered diet history questionnaire (BDHQ) to assess the Japanese diet, which uses food frequency and dietary history. The BDHQ is a validated food frequency questionnaire that estimates the dietary intake of 58 food and beverage items during the preceding month. It consists of the following five sections: intake frequency of food and non-alcoholic beverages, daily intake of rice and miso soup, frequency of drinking alcoholic beverages and amount per drink for five alcoholic beverages, usual cooking methods, and general dietary behaviors [35,36,37]. The food and beverage items included in the questionnaire were mainly from a food list used in Japan′s National Health and Nutrition Survey, which is based on foods commonly consumed in Japan [35].

It is widely established that there are relationships between nutrition and hypertension [38,39,40], and between oral health and hypertension [7,8,9,10,41]. A study by Fushida et al. [41] elucidated the link between high BP and decreased masticatory performance; however, their study did not assess the role of oral health in nutrition. As stated by the authors, a non-direct causal relationship was assumed between high BP and decreased masticatory ability [41]. Nutritional status is expected to be very strongly associated with the relationship between high BP and decreased masticatory performance, among several other expected confounding factors [8,41]. Therefore, the authors felt the need to investigate further the association of oral health with hypertension, indirectly, by assessing nutrition, to better understand the cardiovascular demographics in older adults.

The authors hypothesized that impaired oral health can cause nutritional imbalances, which might affect blood pressure. Hence, in this study, we aimed to investigate the dietary intake of the Japanese older adult population to clarify the role of oral health in nutrition and hypertension.

## 2. Materials and Methods

### 2.1. Study Design and Research Subjects

This prospective cross-sectional study was approved by the Institutional Review Board of Hyogo Medical College (approval number: Rinhi 0342), and it was also part of the frail elderly study in the Tamba Sasayama-Area (FESTA), a cohort study conducted in Tamba Sasayama-Area Hyogo Prefecture, Japan. The study population was composed of healthy community-dwelling older adult individuals aged 65 years and above who voluntarily participated and provided written informed consent in a joint medical and dental study between April 2016 and December 2019.

Research participants were recruited through advertising in local newspapers and posters at the Hyogo College of Medicine Sasayama Medical Center. Participants included in this FESTA study were independent older adult who needed less than level 1 care in the long-term care insurance system in Japan. The exclusion criteria were participants suspected of having moderate to severe dementia (Mini-Mental Condition Screening score < 20). Participants were informed on the aim and method of the study before beginning the comprehensive assessments.

We recruited 921 participants (the baseline data) with complete anthropometric and BDHQ data, while we excluded six participants without oral health assessment data. Based on the BDHQ data, we also excluded 21 participants with a calorie intake of <600 kcal/day or >4000 kcal/day (<600 kcal/day is equivalent to half the calorie intake required for the lowest physical activity while >4000 kcal/day is equivalent to 1.5 times the energy intake required for medium physical activity) [42] because extremely low or high calorie intake values were suspected to be an improper response in the BDHQ survey [36]. After excluding the potential outliers, we enrolled 894 participants aged 65 years and above (mean age: 74.3 ± 5.8 years, 282 men and 612 women) in this study (Figure 1).

### 2.2. Blood Pressure Assessment

Blood pressure measurement was performed between 10:00 and 13:00. Participants were allowed to rest for 10 min prior to the measurement to avoid recording incorrect data. Systolic blood pressure (SBP) and diastolic blood pressure (DBP) were measured twice during daytime (with a 1-minute interval per measurement) using a fully automatic calibrated oscillometer (BP-203 RVII, Colin Co., Aichi, Japan) with an upper-arm cuff device (the participant’s arm was resting on the table during the measurement). The average of two measurements was considered [5].

According to the criteria of the Japanese Society of Hypertension (JSH) 2019 guidelines for the management of hypertension, participants with SBP < 140 mmHg/DBP < 90 mmHg and SBP ≥ 140 mmHg/DBP ≥ 90 mmHg or those found using hypertensive medication are classified as normal and hypertensive, respectively [5,6]. Based on this, we grouped the participants as “Normotensive” (SBP of <140 mmHg and DBP < 90 mmHg), “Hypertensive” (SBP of ≥140 mmHg and/or DBP ≥ 90 mmHg), and “History of hypertension” (history of hypertension in the medical interview and/or is taking hypertension medication). The medications that the participants in the History of hypertension group were taking are shown in Appendix A Table A1 and Table A2.

The difference between the SBP and DBP was recorded as the pulse pressure (PP). PP indicates large-artery stiffness, which is a blood pressure indicator for chronic heart disease risk and advanced atherosclerosis [5,43].

### 2.3. Survey Questionnaire

We asked the participants to complete a survey questionnaire collecting information about their sociodemographic characteristics such as age, sex, education, and living arrangements. Education was defined by the number of years for completion based on the Japanese education system [44], while the living arrangement was defined as alone or with family if more than one family member. Information on health included exercise and smoking habits, and chronic diseases such as diabetes mellitus, dyslipidemia, chronic kidney disease, cardiovascular disease, and stroke were collected from the participants.

### 2.4. Physical Component Assessments

We determined the participants’ body mass index (BMI), skeletal muscle mass, body fat mass, and percent body fat using a body composition analyzer InBody770 (InBody, Tokyo, Japan) with a bio-electrical impedance. The BMI (kg/m^2^) was defined as the weight divided by the square of the height. Skeletal muscle mass was the summation of the mass of the upper and lower extremities [45,46], body fat mass was the summation of fat at the surface level and internal fat, and percent body fat, an indicator of the risk of obesity, was body fat mass divided by total weight [47].

### 2.5. Oral Health Assessment

Assessment of oral health was conducted by calibrated dental examiners. The examinations were performed under bright artificial lighting while the participants sat on a reclining care chair. Oral health was defined on the basis of the number of remaining teeth and oral functions [13,48,49]. In this study, we determined the number of remaining teeth (remaining root fragments and third molars; min, max: 0, 32) [49] and oral functions (maximum occlusal force, posterior occlusal contact, masticatory performance, oral moisture, and oral bacterial counts). To determine the maximum occlusal force, the participants were asked to bite an Occlusal Force-Meter GM10 (Nagano Keiki, Tokyo, Japan). When one of the first molars was missing, the participants were asked to bite on the closest teeth based on the location of the missing first molar [48,50]. Denture-wearing participants were asked to wear their dentures during the measurements. Masticatory performance (MP) assessment was performed using a test gummy jelly (UHA Mikakuto Co., Ltd., Osaka, Japan). Participants were instructed to chew the gummy jelly 30 times and expectorate all the chewed gummy particles on top of a gauze spread over a paper cup. Masticatory performance was evaluated using the visual scoring method on a 10-point scale (0 = minimum to 9 = maximum) [51]. Posterior occlusal contact was defined as the tooth with an occluding antagonist, which may be a natural dentition or a fixed denture [15,21]. According to Eichner’s classification, the occlusal contact in each of the premolar and molar regions was classified into group A (occlusal contact in all four occlusal contact zones), group B (occlusal contact in one to three occlusal contacting zones), and group C (no antagonist contacts in the dentition) [15,21]. In this study, the participants were divided into three groups according to the availability of posterior teeth as the posterior occlusal contact; Eichner A1, A2, A3, B1, B2, and B3 (Figure 2A, “with posterior occlusion (w/PO) group“), Eichner B4, C1, and C2 (Figure 2B, “without posterior occlusion (w/o PO) group“), and Eichner C3 (Figure 2C, “edentulous group”) [19].

Oral moisture was assessed twice by measuring the wetness on the dorsum of the tongue and buccal mucosa using an oral moisture meter (Mucus^®^, LIFE Co., Ltd., Saitama, Japan). A dielectrophoretic impedance measurement method (Panasonic Healthcare Co., Tokyo, Japan) was used to evaluate the bacterial count on the tongue surface [54,55]. The machine rates bacterial counts from levels 1 to 7 [56]; level 1, 2, 3, 4, 5, 6, and 7 indicate the bacterial counts of <10^5^, ≥10^5^ and <10^6^, ≥10^6^ and <3.16 × 10^6^, ≥3.16 × 10^6^ and <10^7^, ≥10^7^ and <3.16 × 10^7^, ≥3.16 × 10^7^, <10^8^, and ≥10^8^, respectively [49].

### 2.6. Nutrition Assessment Methods

Dietary intakes were evaluated using a brief-type self-administered diet history questionnaire (BDHQ), a previously validated fixed-portion type food frequency questionnaire [37]. This questionnaire explores the general dietary habits, cooking methods, and intake frequency of 58 foods and beverages consumed in Japan, including the daily intake of rice and miso soup, consumption frequency of non-alcoholic beverages, and the amount per drink consumed for five alcoholic beverages [36]. In this study, we asked the participants about the consumption frequency of selected foods during the previous month, without mentioning portion size, while a managerial dietician or investigator helped them complete the questionnaire. The food groups included in the BDHQ were meat, fish, vegetables, fruits, cereals, seasonings/condiments, fermented soybean paste (miso), noodle soup, confectionaries, alcoholic and non-alcoholic beverages, and dairy products. The participants were asked about the consumption frequency of each food (once a day, twice or more daily, once a week, 2–3 times a week, 4–6 times a week, less than once a week, or did not eat/drink). Using the responses for BDHQ and an ad hoc computer algorithm based on the Standard Tables of Food Composition in Japan, the daily intake of food items, mean daily intake of energy, and chosen nutrients were determined [35,36,37]. High sodium intake was reported to be associated with hypertension; however, potassium intake or sodium-to-potassium ratio must also be considered [57]. Hence, we calculated the sodium-to-potassium ratio from the quantities of sodium and potassium intake from the BDHQ data [40]. Although this has not been validated yet, previous studies have reported that the high potassium intake or low sodium-to-potassium ratio may have beneficial possibilities for BP [57]. Daily alcohol consumption was calculated as a part of the BDHQ [35,37]. The data used in this study were coded to preserve the anonymity of the participants.

### 2.7. Sample Size Calculation and Statistical Analysis

This was a prospective cohort study, and the sample size was calculated based on the data of previous studies that assessed the relationship between dietary intakes and oral functions [22,36]. Since sodium intake was found to be the factor closely related to the BP of older adults in a previous study [5], we found it appropriate to determine the sample size of this study based on the sodium intake. Based on this, the minimum number of participants required in each group was 185, and, hence, the analysis was performed with data, including those of the participants who presented to our study until December 2019 (at this time, the number of participants in each group exceeded the required number as per the sample size calculation).

The normality of the data distribution was verified using the Kolmogorov–Smirnov test (*p* > 0.05); the data were found to be non-normally distributed. To determine the differences in oral health and dietary intake among the participants with different BP groups, Kruskal-Wallis and Chi-square tests were performed, while intergroup comparisons were performed using the Mann-Whitney U-test with Bonferroni correction. The measurement items were specified according to a Kruskal-Wallis or X^2^ test (*p* < 0.05, explanatory variable for basic characteristics and oral health items; *p* < 0.1, explanatory variable for dietary intake). We used the Spearman’s rank correlation coefficient to examine the relationship between oral health and dietary intake. To determine the impact of hypertension-related factors, we performed binary logistic regression analyses with stepwise methods (input 0.05, remove 0.15). Hypertension, defined as participants classified as Hypertensive or History of hypertension, was considered the objective variable (Normotensive participants = 0, Hypertensive or History of hypertension participants = 1). The explanatory variables were selected from the participants’ basic characteristics, oral health, and dietary intake, which were found to be significantly related to hypertension by the Kruskal-Wallis or chi-square test. All statistical analyses were performed using the SPSS Statistics 22.0 (IBM, Tokyo, Japan).

## 3. Results

### 3.1. Subject Characteristics

Table 1 provides an overview of the participants. This study included 894 participants (mean age: 74.3 ± 0.2 years old), most of whom were female (68.5%). The results also indicated that 45.3% of the participants had history of hypertension, with females accounting for 64.4% of the History of hypertension group. Overall, the aggregate prevalence of hypertension in the Hypertensive and the History of hypertension groups was 69.1%, which was relatively high. Among the three groups, statistically significant differences were found in sex, smoking habits, SBP and DBP, pulse pressure, cardiovascular disease, and stroke. In contrast, we only found significant differences between the Normotensive and the History of hypertension groups in terms of age. The Normotensive group was statistically different from the Hypertensive and the History of hypertension groups in terms of BMI, body fat mass, and percent body fat, while a significant difference between the Hypertensive and the History of hypertension groups was observed for the skeletal muscle mass. No significant differences were found in education, living arrangement, exercise habits, alcohol consumption, diabetes mellitus, dyslipidemia, and chronic kidney disease.

### 3.2. The Relationship between Oral Health and Hypertension

Table 2 shows the analysis of oral health parameters and their association with hypertension. The mean average of the remaining teeth among participants in the History of hypertension group was significantly lower than that of the Normotensive group, while the Normotensive group had significantly higher masticatory performance than that of the Hypertensive or the History of hypertension groups. It was found that there was a significant difference in posterior occlusal contact between the Normotensive and the History of hypertension groups. Among all the participants, 17.9% had no posterior occlusal contact, with the History of hypertension group having the highest prevalence of having no posterior occlusal contact. Other parameters such as occlusal force, oral moisture, bacterial count, and levels were not associated with BP condition. Although there was no significant difference in oral moisture, it can be noted that the History of hypertension group had the lowest oral moisture count among all the groups.

### 3.3. Comparison of Nutrient and Dietary Intake among Normotensive, Hypertensive, and History of Hypertension Groups

Table 3 shows the comparison between the nutrient and dietary intake among the three groups (Normotensive, Hypertensive, and History of hypertension). There were significant differences in sodium-to-potassium ratio, β-carotene, vitamin K, and meat intake. The sodium-to-potassium ratio of the Normotensive group was significantly lower than that of the other two groups, while intakes of β-carotene, Vitamin K, green and yellow vegetables, and meat were significantly lower in the History of hypertension group. Moreover, it was also found that the sodium-to-potassium ratio (1.80 ± 0.06) of males and the meat intake in the Hypertensive group was highest among the groups (Appendix A Table A3). In other words, participants in the History of hypertension group had a lower intake of green and yellow vegetables, β-carotene, and vitamin K, and higher sodium-to-potassium ratio.

### 3.4. Relationship among Nutrient, Food Items, and Oral Health of the Participants

Although significant differences were found among the three groups, as seen in Table 3, age was the only factor significantly correlated with β-carotene, vitamin K, green and yellow vegetables, and meat. The sodium-to-potassium ratio was found to show a statistically significant correlation between the remaining teeth and masticatory performance; however, the correlation was weak (Table 4).

### 3.5. Factors Affecting Hypertension

The logistic regression analysis revealed that age, BMI, sodium-to-potassium ratio, and absence of posterior occlusal contact were significant explanatory variables affecting hypertension and/or need for BP control with medication in the participants of this study (Table 5). The risk of hypertension and/or need for BP control with medication increased with age (odds ratio (OR): 1.04; confidence interval (CI): 1.01, 1.07), it was 1.2 times higher in participants with a higher BMI (OR: 1.23; CI: 1.16, 1.30), 1.7 times in those with a higher intake of sodium-to-potassium ratio (OR: 1.65; CI: 1.12, 2.43), and 1.7 times in those without posterior occlusal contact (OR: 1.73; CI: 1.11, 2.69).

## 4. Discussion

In this study, we aimed to assess the role of oral health in nutrition and hypertension by exploring dietary intakes in a Japanese older adult population. Impaired mastication makes an individual more susceptible to developing systemic diseases that could lead to frailty and mortality when not addressed properly. Among the participants in this study, 45.3% had a history of hypertension, with the majority being females (64.4%), older, and having a higher BMI, higher alcohol consumption, exercise and smoking habits, comorbidity diseases such as diabetes mellitus, dyslipidemia, chronic kidney disease, cardiovascular disease (CVD), stroke, having a lower number of remaining teeth, loss of posterior occlusal contact, decreased occlusal force, lower oral moisture count, lower masticatory performance, lower intake of green and yellow vegetables, β-carotene, vitamin K, and high sodium-to-potassium ratio. This study showed that age was the only factor significantly associated with β-carotene, vitamin K, green and yellow vegetables, and meat, while the sodium-to-potassium ratio barely correlated with the remaining teeth and masticatory performance. Furthermore, significant explanatory variables affecting hypertension among the participants were BMI, age, sodium-to-potassium ratio, and the absence of posterior occlusal contact. According to prior findings, periodontal disease can affect the number of remaining teeth [10]. Decrease in posterior occlusal contact can lead to a decline in mastication ability [17]. Since mastication is an essential function for ingestion and digestion, a decline in mastication ability induces many functional declines, including malnutrition. If problems pertaining to nutrition are not addressed appropriately, it could lead to systemic issues, such as hypertension [17]. The results of our study postulate that decreased posterior occlusal contact possibly restricts individuals to chew foods, such as green and yellow vegetables, thus consequently decreasing dietary fiber intake, leading to low potassium and high sodium intake, which are risk factors for developing hypertension. The study findings suggest that maintaining good oral health is important to improve nutritional intake and prevent hypertension. To our knowledge, this is the first study that acknowledged oral health as another contributing factor in the development of high BP by examining the dietary intakes of the older adult in the Japanese population.

### 4.1. The Role of Oral Health in Nutrition and Hypertension

The findings of this study revealed that the risk of hypertension increased with age, which was consistent with previous reports [5,16]. The study also revealed that in older adults, the absence of posterior occlusal contact increased the risk of hypertension by 1.7 times. The loss of occlusal contact results in a decrease in mastication ability [17]. Thus, this study postulates that the participants in the History of hypertension group were found to have a lower masticatory performance because they had lower oral moisture count, decreased occlusal force, no posterior occlusal contact, and a lesser number of remaining teeth. Tooth loss can lead to hyperactivity of the masticatory system, leading to unhealthy diet patterns and reduced nutrient intake that can negatively influence the general health [58]. Studies have reported that tooth loss was directly related to reduced oral function [17]. Hence, the Japan Ministry of Health, Labor, and Welfare continuously promotes preserving ≥ 20 natural teeth until the age of 80 [58] because tooth loss can decrease posterior occlusal contact, consequently influencing the occlusal force [59]. Reduced occlusal force affects saliva secretion, which is necessary for oral functions such as mastication and swallowing. Other factors such as systemic diseases and medications can influence saliva secretion [17,60]. Hasegawa et al., found that the patients taking antihypertensive drugs have decreased saliva secretion [60]. Dry mouth is a common side effect of antihypertensive medication, which could deteriorate teeth and periodontal condition [49], eventually leading to tooth loss if not addressed properly. In this study, the participants were found to be taking calcium channel blockers (CCBs), renin–angiotensin system inhibitors (angiotensin II receptor blockers (ARBs), angiotensin-converting enzyme (ACE) inhibitors), aldosterone receptor antagonists, β-blockers, and α-blockers. Moreover, there were several participants who took two combinations of medication. It was reported that the masticatory ability was influenced by the number of natural teeth, occlusal support, muscular weakness, and occlusal force [23].

Mastication is a necessary function for nutrition-ingestion, and an impaired mastication ability leads to many functional declines, including malnutrition [17]. Several observational studies have revealed that tooth loss, particularly in older adults, was associated with dietary intake of fruits and vegetables and a low intake of antioxidative vitamins. Furthermore, the low occlusal force has been linked to a decrease in vegetable, fruit, antioxidant, and dietary fiber intake [23]. The present study presumed that the individuals without posterior occlusal contact prefer foods that are easy to chew and avoid vegetables high in potassium, which was consistent with a study finding that Japanese older adults with <19 teeth had a significantly lower intake of vegetables than those with ≥20 [61]. This was also consistent with the findings of our study that the participants in the History of hypertension group had the lowest intake of potassium. According to the JSH 2019 guidelines, improving dietary patterns, such as increasing fruit and vegetable intakes (to 350 g/day), can reduce the risk of hypertension because potassium antagonizes the hypertensive activity of the sodium [5]. It was also pointed out that balanced ingestion of fruits and vegetables (especially green and yellow vegetables) may be necessary for improved general health [17].

### 4.2. Other Factors Related to Hypertension

Our findings also revealed that the participants with a higher intake of sodium-to-potassium ratio had 1.7 times higher risk of developing hypertension, which confers with the report of Park et al. [40] that sodium-to-potassium ratio and blood pressure were strongly correlated. Moreover, participants in the Hypertensive group had the highest intake of sodium (Na) and salt (NaCl), 5082.4 mg/day and 12.8 g/day, respectively, which was higher than the Japanese Society of Hypertension′s recommendation of 6.0 g/day for hypertensive individuals [5]. However, because the BDHQ was designed to evaluate Japanese dietary habits and was not specific for sodium intake, we were unable to establish the relationship between salt intake, sodium intake, and hypertension in this study [57]. The current study considered that the easy-to-chew foods preferred by the older Japanese adults were processed foods with high salt content [5]. It was previously reported that the high salt intake increases BP, and excessive salt intake was one of the possible causes for the high prevalence of hypertension and stroke in Japan [5,62]. Furthermore, we also found that even if the participants in this study had a normal BMI, their BP was still higher, which was consistent with the report of the JSH 2019 that the hypertensive Japanese are often free of obesity [5]. The results of our study also suggested that the participants with a higher BMI were 1.2 times more likely to develop hypertension and/or need BP control, which was also agreeing with the JSH 2019 report that population risk for cardiovascular diseases was higher in hypertensive non-obese individuals than that in the hypertensive obese individuals [5].

Hence, the Japanese government conducted mass media-mediated public education, obligated food manufacturers to indicate salt content in food packaging, promoted nutritional labeling in school lunch/food service industries, distributed home blood pressure measurements, and required all allied health professionals to instruct patients including non-hypertensive individuals to improve their lifestyle including the balanced dietary intake, improved oral hygiene, increased physical activities (approximately 1500 step increase in the number of steps), and maintain moderate alcohol consumption. Furthermore, the Japanese government provided home-visit dental services to promote the oral health of dependent older adults and covered dental care as part of its universal health coverage [5,13]. These strategies are required to achieve the goals of reducing hypertension, CVD morbidity/mortality, and extending the healthy life expectancy of Japanese individuals.

### 4.3. Limitation

Several limitations must be considered when interpreting the findings of this study. First, there might have been an underestimation or overestimation of dietary intake because the BDHQ is a self-report survey designed to evaluate Japanese dietary patterns, habitual intake, cooking, and seasoning, which did not reflect the quantity of selected foods [63]. Second, the sodium-to-potassium ratio was not calculated using the urine analysis [57]. Third, we failed to find a relationship between education and living arrangements with the hypertension risk because highly educated individuals may be more knowledgeable about choosing healthy food [64], while individuals who eat alone might choose a quick and simple meal rather than a nutritionally balanced diet [65]. This might happen because we evaluated them based on their educational years rather than the educational level. Lastly, no causal relationships among dietary intake, oral health, and hypertension were established because of the observational design of the study. Hence, further studies should be conducted in the future to elucidate more on the role of oral health in nutrition.

### 4.4. Clinical Implications

Given the significance of oral health in the nutritional status and general health of older adults, preserving remaining teeth and improving masticatory performance is critical. This study provided evidence that decreased posterior occlusal contact increases the risk of hypertension by 1.7 times. Hence, oral rehabilitation with dentures to maintain posterior occlusal contact may be recommended for improved masticatory performance, improved dietary intake, and a lower risk of hypertension. The findings of this study will also raise awareness about lifestyle changes that can reduce the risk of hypertension, such as lower salt intake, higher fruit and vegetable intake, maintaining a healthy weight, increasing exercise habits, decreasing smoking habits, and reducing alcohol consumption.

## 5. Conclusions

This study suggests that the risk of hypertension increased with age, higher BMI, higher sodium-to-potassium ratio, and lesser posterior occlusal contact. Therefore, maintaining posterior occlusal contact is vital in lowering the risk of hypertension. The findings of this study will aid in the improvement of oral health, nutritional intake, and general health by reducing the development of hypertension and potentially extending the healthy life expectancy of older adults.

## Figures and Tables

**Figure 1 nutrients-14-01279-f001:**
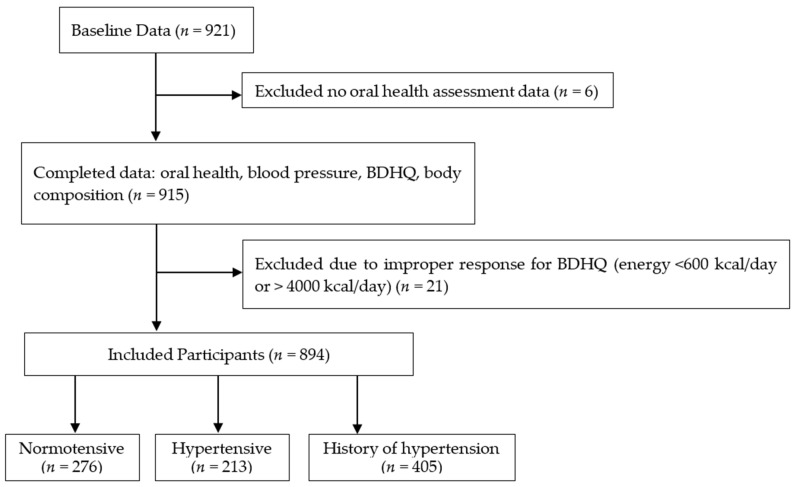
Flow diagram demonstrating the recruitment and group assignment of the study participants. BDHQ; a brief self-administered diet history questionnaire. Normotensive: Participants with systolic blood pressure (SBP) < 140 mmHg and diastolic blood pressure (DBP) < 90 mmHg, Hypertensive: Participants with SBP of ≥140 mmHg and/or DBP ≥ 90 mmHg, History of hypertension: Participants who answered that they had a history of hypertension in the medical interview and/or is taking hypertension medication.

**Figure 2 nutrients-14-01279-f002:**
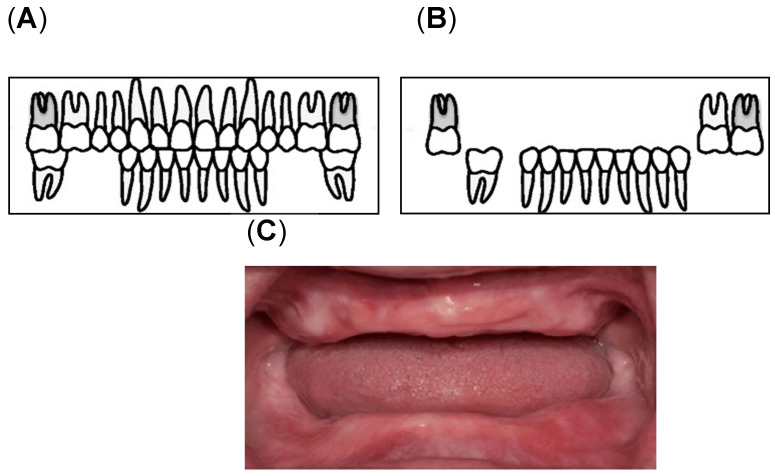
Classification of the Posterior Occlusal Contact: (**A**) With posterior occlusion (with PO) group consisting of Eichner A1, A2, A3, B1, B2, and B3; (**B**) Without Posterior occlusion (without PO) group consisting of Eichner B4, C1, and C2 [52]; (**C**) the edentulous group [53].

**Table 1 nutrients-14-01279-t001:** Subject characteristics.

Measurement Variables	Total (*n* = 894)	Normotensive Group (*n* = 276)	Hypertensive Group (*n* = 213)	History of Hypertension Group (*n* = 405)	*p*-Value	Two-Group Comparison
Age (Years) *	74.3 ± 0.2	73.3 ± 0.3	74.3 ± 0.4	75.0 ± 0.3	0.002	b
Sex **						
Male	282 (31.5)	82 (29.7)	56 (26.3)	144 (35.6)	0.046	c
Female	612 (68.5)	194 (70.3)	157 (73.7)	261 (64.4)		
Education (years)	12.4 ± 0.1	12.4 ± 0.1	12.4 ± 0.1	12.3 ± 0.1	0.80	
Living arrangement						
Alone	98 (14.1)	33 (15.1)	24 (14.5)	41 (13.3)	0.83	
With Family	596 (85.9)	186 (84.9)	142 (85.5)	268 (86.7)		
Exercise habits	524 (58.7)	178 (64.5)	120 (56.3)	226 (55.8)	0.06	
Exercise Frequency (times)	14.1 ± 0.5	14.4 ± 0.8	12.6 ± 0.9	14.7 ± 0.7	0.29	
Exercise periods (years)	11.4 ± 0.7	10.9 ± 1.1	11.8 ± 1.5	11.7 ± 1.2	0.88	
Alcohol Consumption (g/day)	7.3 ± 0.5	6.0 ± 15.7	5.9 ± 0.9	8.8 ± 0.9	0.32	
Current or past smoking habit **	247 (27.6)	71 (25.7)	48 (22.5)	128 (31.6)	0.039	
Blood Pressure						
Systole (mmHg) *	139.3 ± 0.6	126.8 ± 0.6	152.8 ± 0.9	140.7 ± 0.9	<0.001	a b c
Diastole (mmHg) *	80.0 ± 0.4	73.5 ± 0.5	87.0 ± 0.6	80.8 ± 0.5	<0.001	a b c
Pulse Pressure (mmHg) *	59.3 ± 0.4	53.4 ± 0.5	65.8 ± 0.7	59.9 ± 0.6	<0.001	a b c
Body Composition						
BMI (kg/m^2^) *	22.7 ± 0.1	21.7 ± 0.2	22.8 ± 0.2	23.3 ± 0.1	<0.001	a b
Skeletal Muscle Mass (kg) *	21.0 ± 0.1	20.6 ± 0.2	20.4 ± 0.3	21.6 ± 0.2	0.003	b c
Body Fat Mass (kg) *	15.4 ± 0.2	13.9 ± 0.3	15.8 ± 0.4	16.1 ± 0.3	<0.001	a b
Percent Body Fat *	27.8 ± 0.3	26.3 ± 0.4	28.8 ± 0.5	28.4 ± 0.4	<0.001	a b
Comorbidity Disease						
Diabetes mellitus	107 (12.0)	29 (10.5)	20 (9.4)	55 (13.6)	0.55	
Dyslipidemia	208 (23.3)	58 (21.0)	51 (23.9)	99 (24.4)	0.56	
Chronic kidney disease	27(3.0)	5 (1.8)	8 (3.8)	14 (3.5)	0.36	
Cardiovascular disease (CVD) **	61 (6.8)	17 (6.2)	7 (3.3)	37 (9.1)	0.020	
Stroke **	12 (1.3)	1 (0.4)	1 (0.5)	10 (2.5)	0.029	

Data are presented as mean ± standard error or number of participants (%). The significance level was set at *p* < 0.05. *: Significant differences were seen using the Kruskal-Wallis Test, **: Significant differences were seen using the X^2^-test. BMI: body mass index; Normotensive: Participants with systolic blood pressure (SBP) < 140 mmHg and diastolic blood pressure (DBP) < 90 mmHg; Hypertensive: Participants with SBP of ≥140 mmHg and/or DBP ≥ 90 mmHg; History of hypertension: Participants who answered that they had a history of hypertension in the medical interview and/or is taking hypertension medication; Exercise habits: participants who regularly exercise; exercise frequency: the frequency of exercise every month; exercise periods: the number of years the participants have been exercising. Current or past smoking habits: Participants who smoked or had a smoking history. The Mann-Whitney U-test with Bonferroni correction was used to compare the two groups: a: statistically significant difference between the Normotensive and the Hypertensive groups; b: statistically significant difference between the Normotensive and History of hypertension groups; c: statistically significant difference between the Hypertensive and History of hypertension groups.

**Table 2 nutrients-14-01279-t002:** The Relationship Between Oral Health and Hypertension.

Variables	Total (*n* = 894)	Normotensive Group (*n* = 276)	Hypertensive Group (*n* = 213)	History of Hypertension Group (*n* = 405)	*p*-Value	Two-Group Comparison
Remaining teeth (teeth) *	19.8 ± 0.3	21.2 ± 0.5	19.6 ± 0.6	19.1 ± 0.5	0.007	b
Maximum Occlusal force (kgf)	51.1 ± 1.3	53.5 ± 2.3	51.4 ± 2.5	49.2 ± 1.9	0.27	
Masticatory Performance (score) *	4.0 ± 0.1	4.3 ± 0.1	3.7 ± 0.2	3.9 ± 0.1	0.004	a b
Posterior occlusal contact **					0.012	b
With posterior occlusion (w/PO)	671 (75.1)	226 (81.9)	159 (74.6)	286 (70.6)		
Without posterior occlusion (w/o PO)	160 (17.9)	32 (11.6)	39 (18.3)	89 (22.0)		
Edentulous	63 (7.0)	18 (6.5)	15 (7.0)	30 (7.4)		
Oral Moisture						
Cheek	28.4 ± 0.2	28.5 ± 0.2	28.6 ± 0.7	28.2 ± 0.2	0.42	
Tongue	26.8 ± 0.1	27.1 ± 0.2	26.6 ± 0.3	26.8 ± 0.2	0.45	
Total mucus	55.2 ± 0.3	55.6 ± 0.4	55.2 ± 0.9	55.0 ± 0.4	0.43	
Bacterial count (CFU/mL)	4.6 × 10^7^ ± 1.7 × 10^7^	6.6 × 10^7^ ± 4.3 × 10^7^	2.3 × 10^7^ ± 3 × 10^6^	4.6 × 10^7^ ± 2.3 × 10^7^	0.83	
Bacterial level (min. max: 1. 7)	4.8 ± 0.03	4.8 ± 0.1	4.8 ± 0.1	4.8 ± 0.04	0.97	

Data are presented as mean ± standard error or number of participants (%). The significance level was set at *p* < 0.05. * Significant differences were seen using the Kruskal-Wallis test; **: Significant differences were seen using the X^2^-test. kgf: kilogram-force, CFU: colony-forming unit, min: minimum, max: maximum. Posterior occlusal contact was classified into three groups according to the availability of posterior teeth based on the Eichner index. With posterior occlusion (with/PO) group consisting of Eichner A1, A2, A3, B1, B2, and B3; Without Posterior occlusion (w/o PO) group consisting of Eichner B4, C1, and C2; the edentulous group consisting of Eichner C3 [19]. The Mann-Whitney U-test with Bonferroni correction was used to compare the two groups: a: statistically significant difference between the Normotensive and the Hypertensive groups; b: statistically significant difference between the Normotensive and History of hypertension groups; c: statistically significant difference between the Hypertensive and History of hypertension groups.

**Table 3 nutrients-14-01279-t003:** Comparison of nutrient and dietary intake among groups.

Variables	Total (*n* = 894)	Normotensive Group (*n* = 276)	Hypertensive Group (*n* = 213)	History of Hypertension Group (*n* = 405)	*p*-Value	Two-Group Comparison
Nutrient Intake						
Energy (kcal/day)	2083.8 ± 21.4	2080.1 ± 38.0	2099.6 ± 46.2	2078.0 ± 31.1	0.97	
Carbohydrates (g/day)	263.6 ± 2.9	263.8 ± 5.2	265.2 ± 6.1	262.6 ± 4.3	0.99	
Protein Total (g/day)	90.8 ± 1.2	91.1 ± 2.1	92.5 ± 2.6	89.8 ± 1.7	0.93	
Animal	56.5 ± 0.9	56.0 ± 1.6	58.2 ± 2.1	56.1 ± 1.4	0.96	
Vegetable	34.3 ± 0.4	35.1 ± 0.7	34.3 ± 0.8	33.8 ± 0.5	0.47	
Fat Total (g/day)	66.1 ± 0.8	66.6 ± 1.4	67.4 ± 1.8	65.1 ± 1.2	0.65	
Animal	32.3 ± 0.5	32.3 ± 0.8	33.1 ± 1.1	31.8 ± 0.7	0.83	
Vegetable	33.9 ± 0.4	34.4 ± 0.8	34.3 ± 0.9	33.3 ± 0.6	0.59	
Sodium (mg/day)	4994.3 ± 59.5	4950.0 ± 103.6	5082.4 ± 130.4	4978.3 ± 87.1	0.95	
Potassium (mg/day)	3392.7 ± 44.0	3506.1 ± 81.0	3382.5 ± 92.9	3320.7 ± 63.0	0.34	
Sodium-to-Potassium Ratio *	1.55 ± 0.01	1.48 ± 0.02	1.56 ± 0.03	1.58 ± 0.02	0.002	a b
Calcium (mg/day)	806.0 ± 11.5	828.9 ± 21.2	797.2 ± 24.0	795.0 ± 16.7	0.39	
Magnesium (mg/day)	329.4 ± 4.1	337.4 ± 7.5	329.4 ± 8.7	323.9 ± 5.8	0.51	
Fatty Acids (g/day)						
Saturated	17.8 ± 0.2	18.0 ± 0.4	18.2 ± 0.5	17.5 ± 0.3	0.51	
Monounsaturated	23.2 ± 0.3	23.3 ± 0.5	23.8 ± 0.6	22.8 ± 0.4	0.56	
Polyunsaturated	15.7 ± 0.2	15.9 ± 0.3	15.9 ± 0.4	15.5 ± 0.3	0.79	
Dietary Fiber (g/day)	15.6 ± 0.2	16.2 ± 0.4	15.6 ± 0.4	15.2 ± 0.3	0.33	
Water-Soluble	4.0 ± 0.1	4.1 ± 0.1	3.9 ± 0.1	3.9 ± 0.1	0.29	
Insoluble	11.1 ± 0.1	11.5 ± 0.3	11.1 ± 0.3	10.8 ± 0.2	0.29	
β-carotene (μg/day) *	5331.5 ± 103.5	5760.4 ± 206.3	5251.8 ± 200.0	5081.2 ± 145.1	0.07	b
Retinol (mg/day)	1064.4 ± 24.8	1115.0 ± 45.7	1015.2 ± 50.1	1055.3 ± 36.5	0.41	
Vit. B1 (mg/day)	1.0 ± 0.0	1.0 ± 0.0	1.0 ± 0.0	1.0 ± 0.0	0.55	
Vit. B2 (mg/day)	1.7 ± 0.0	1.8 ± 0.0	1.7 ± 0.0	1.7 ± 0.0	0.24	
Niacin (mg/day)	21.8 ± 0.3	21.9 ± 0.5	22.1 ± 0.7	21.5 ± 0.5	0.87	
Vit. B6 (mg/day)	1.7 ± 0.0	1.7 ± 0.0	1.7 ± 0.0	1.6 ± 0.0	0.73	
Vit. B12 (μg/day)	14.5 ± 0.3	14.1 ± 0.5	14.5 ± 0.6	14.8 ± 0.5	0.78	
Folate (μg/day)	470.9 ± 6.5	492.0 ± 11.9	466.4 ± 13.3	458.8 ± 9.4	0.12	
Pantothenic Acid (mg/day)	8.3 ± 0.1	8.4 ± 0.2	8.3 ± 0.2	8.2 ± 0.1	0.75	
Vit. C (mg/day)	167.3 ± 2.7	172.0 ± 4.9	167.8 ± 5.5	163.7 ± 4.0	0.51	
Vit. D (μg/day)	24.6 ± 0.6	24.3 ± 1.0	25.1 ± 1.3	24.6 ± 0.9	0.99	
Vit. K (μg/day) *	425.1 ± 7.6	451.8 ± 14.3	425.5 ± 15.8	406.7 ± 10.8	0.07	b
Food Groups (g/day)						
Cereals	371.2 ± 5.5	367.1 ± 9.5	374.9 ± 11.1	372.1 ± 8.6	0.85	
Potatoes	65.8 ± 1.9	68.2 ± 3.6	62.4 ± 3.7	66.0 ± 2.9	0.59	
Sugars and sweeteners	6.1 ± 0.1	6.0 ± 0.3	6.4 ± 0.3	5.9 ± 0.2	0.66	
Legume	89.9 ± 1.7	96.1 ± 3.5	86.3 ± 3.5	87.6 ± 2.3	0.23	
Green and yellow vegetables *	143.2 ± 2.9	154.5 ± 5.5	139.8 ± 5.9	137.3 ± 4.1	0.06	b
Other vegetables	221.6 ± 4.0	232.7 ± 7.6	220.3 ± 8.2	214.6 ± 5.7	0.21	
Fruits	162.1 ± 4.1	157.3 ± 6.7	165.0 ± 7.9	163.9 ± 6.5	0.74	
Seafood	125.5 ± 3.0	122.1 ± 5.0	128.5 ± 6.5	126.3 ± 4.4	0.90	
Meat *	80.6 ± 1.7	79.1 ± 2.9	88.9 ± 3.8	77.3 ± 2.4	0.03	c
Eggs	49.9 ± 1.1	50.2 ± 1.8	48.8 ± 2.0	50.3 ± 1.7	0.76	
Dairy Products (Milk)	184.4 ± 4.0	194.9 ± 7.7	172.7 ± 8.0	183.5 ± 5.7	0.24	
Fats and oils	11.5 ± 0.2	11.4 ± 0.3	11.9 ± 0.4	11.3 ± 0.3	0.48	
Confectionery	69.3 ± 1.9	68.6 ± 3.4	71.3 ± 4.1	68.7 ± 2.6	0.85	
Beverages	703.6 ± 11.8	737.3 ± 22.1	688.3 ± 24.4	688.7 ± 16.8	0.21	
Condiments	226.3 ± 5.0	224.5 ± 9.4	236.6 ± 10.6	222.1 ± 6.9	0.56	
Salt Intake	12.6 ± 0.2	12.5 ± 0.3	12.8 ± 0.3	12.6 ± 0.2	0.95	

Dietary assessment is the result of a brief self-administered diet history questionnaire (BDHQ). The results are expressed as the mean ± standard error. The significance level was set to *p* < 0.1. The sodium-to-potassium ratio was defined as the amount of sodium consumed divided by the amount of potassium consumed [40,57], Vit.: Vitamin. * Significant differences were observed using the Kruskal-Wallis test. Mann-Whitney′s U-test with Bonferroni correction was used to compare the two groups: a: statistically significant difference between the Normotensive and the Hypertensive group; b: statistically significant difference between the Normotensive and the History of hypertension group; c: statistically significant difference between the Hypertensive group and the History of hypertension group.

**Table 4 nutrients-14-01279-t004:** Relationship among nutrient, food items, and oral health.

Variables	Age		Remaining Teeth		Masticatory Performance	
Overall	*r*	*p*-Value	*r*	*p*-Value	*r*	*p*-Value
Sodium-to-potassium ratio	0.04	0.23	−0.07 *	0.03	−0.09 *	0.01
β-carotene	0.10 **	0.003	0.04	0.28	0.03	0.46
Vitamin K	0.16 **	<0.001	0.02	0.61	0.03	0.45
Green and yellow vegetables	0.12 **	<0.001	0.03	0.32	0.03	0.38
Meat	0.09 **	0.006	0.02	0.51	0.01	0.82
Normotensive group						
Sodium-to-potassium ratio	0.08	0.16	−0.05	0.36	−0.10	0.09
β-carotene	0.11	0.08	0.01	0.92	0.03	0.63
Vitamin K	0.19 **	0.001	−0.02	0.69	−0.02	0.73
Green and yellow vegetables	0.08	0.17	0.01	0.92	0.05	0.38
Meat	0.20 **	0.001	0.01	0.87	0.00	0.95
Hypertensive group						
Sodium-to-potassium ratio	−0.03	0.62	−0.08	0.26	−0.03	0.66
β-carotene	0.15 *	0.03	0.03	0.68	0.02	0.83
Vitamin K	0.15 *	0.03	0.03	0.67	0.02	0.74
Green and yellow vegetables	0.15 *	0.03	0.05	0.43	0.02	0.74
Meat	0.03	0.61	0.06	0.37	0.07	0.34
History of hypertension group						
Sodium-to-potassium ratio	0.01	0.79	−0.05	0.31	−0.07	0.16
β-carotene	0.09	0.07	0.04	0.43	0.01	0.83
Vitamin K	0.18 **	<0.001	0.02	0.64	0.04	0.40
Green and yellow vegetables	0.15 **	0.002	0.02	0.67	0.00	0.97
Meat	0.05	0.31	0.01	0.89	0.00	0.94

*r*: Spearman correlation coefficient, *: *p* < 0.05, **: *p* < 0.001.

**Table 5 nutrients-14-01279-t005:** Factors affecting hypertension.

Variables	B	Standard Error	Wald	*p*-Value	Odds Ratio	95% CI	
						Lower	Upper
Age (Years) *	0.04	0.02	6.98	0.01	1.04	1.01	1.07
BMI (kg/m^2^) *	0.20	0.03	45.19	<0.001	1.23	1.16	1.30
posterior occlusal contact							
With posterior occlusion (w/PO) *	–	–	6.15	0.05	ref	–	–
Without posterior occlusion (w/o PO) *	0.55	0.23	5.86	0.02	1.73	1.11	2.69
Edentulous	−0.06	0.33	0.03	0.86	0.94	0.49	1.80
Sodium-to-potassium ratio *	0.50	0.20	6.34	0.01	1.65	1.12	2.43
intercept	−7.45	1.31	32.18	<0.001	0.001		

Binary logistic regression analyses with stepwise methods (input: 0.05; removal: 0.15). Nagelkerke R^2^ = 0.125. Only the variables in the equation are presented. Hypertension, defined as participants belonging to the Hypertensive or the History of hypertension groups, was considered the objective variable (normotensive participants = 0, Hypertensive or History of hypertension participants = 1). The variables in Table 1, Table 2 and Table 3, which were significantly associated with hypertension, were entered as explanatory variables. B: Unstandardized coefficient; CI: 95% confidence interval for unstandardized coefficients. * Statistically significant explanatory variables, *p* < 0.1.

## Data Availability

The materials described in the manuscript, including all relevant raw data, will be freely available to any scientist wishing to use them for non-commercial purposes, by contacting the corresponding author without breaching patient confidentiality.

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
