# Peer review of "The Association of Dietary Intake, Oral Health, and Blood Pressure in Older Adults: A Cross-Sectional Observational Study"

_nutrients, 2022, doi:10.3390/nu14061279_

Round 1
Reviewer 1 Report
The study was aimed to investigate the nutrition influence between the oral health and hypertension. The topic is interesting but still, there are some major questions that need to clarify and revise.
Material and methods:
- The lack of any power analysis to determine the appropriate sample size.
- Please explain the methods of detecting the energy level and why the setting value was 4000 kcal/day?
- The time of testing the blood pressure should be mentioned. For example, in the morning or at night?
- What’s the difference of the health condition between normal blood pressure and the participants who with history of hypertension? Will the health condition of the participants who with history was similar with the normal person after they recovery from hypertension? What’s the meaning of collecting the group of HP history group?
- Please put the usage of medicine information from the participants, because the different kinds of medicine might affect the blood pressure.
Results:
- Please add the frequency of the exercise from the participants, due to the number of times and period of exercise that might affect the blood pressure.
- The oral health data including the remaining teeth and posterior occlusion condition, but the lack of oral examination (ex. periodontal status, DMFT….) need to be mentioned. Suggesting recheck the definition of “oral health”, or change to other words.
- Please explain the meaning of “Constant” in Table 5, and its p-value and odds ratio.
- Please explain more about the relationship between oral health data and other influence factors.
Discussion:
- Due to one of the including criteria was BDHQ (energy>4000 kcal/day), but there was no upper limit or lower limit of the value, so is there any possible of the different hypertension or health condition with the participant who’s BDHQ was 5000 kcal/day, but another participant was 15000 kcal/day, which might influence the result? This issue should be discussed.
- The discussion of 4.2., the title was “The relationship of Oral health, Dietary Intake, and Hypertension”, but the part of oral health and the relationship with others were rarely discussed, please discuss more about the oral health and dietary intake (for example, oral moisture effects the diet intake?).
Author Response
RE: (Nutrients-1604969)
Point-by-point responses to the Reviewer 1 comments
Q1. The lack of any power analysis to determine the appropriate sample size.
Author response: We would like to thank the reviewer for mentioning the unclarified issues in our manuscript.
Because this was a prospective cohort study, the number of participants increased during the study period. In a previous study, we have already analyzed and reported the data (collected through 2018) on the association between nutrition and frailty (ref 35, Tamaki et al.). Further, we referred to previous studies to identify the stage for the cross-sectional analysis (ref 21, Iwasaki et al.)
For the three population (changed names; Normotensive, Hypertensive, and History of hypertension), we analyzed a total 168 patients, 84 patients with healthy oral condition (not having impaired dentition) and 84 patients without healthy oral condition (having impaired dentition), to detect an effect size of 0.51 between these groups at a two-sided alpha level of 0.05 and a power of 0.95. Assuming a 10% drop-out level, a total subject of 185 were required. The effect size was calculated, assuming that, for healthy oral condition and non healthy oral condition groups, means were 3.4 and 8.5 with standard deviations of 10 (ref 21, Iwasaki et al). From the above explanation, it is clear that the sample size for this study was sufficient. We have now amended this information in the revised manuscript (Page 6 line number 223-229).
Q2. Please explain the methods of detecting the energy level and why the setting value was 4000 kcal/day?
Author response: We thank you for your comment. The calculations were made using fixed, sex-specific portion sizes of 58 food and beverage items based on food composition list in the Standard Tables of Food Composition in Japan and the energy level was calculated using an ad hoc computer algorithm for the BDHQ (ref 34 and 36, Kobayashi et al.). We have now added this information in the revised manuscript (Page 3 line number 119-124 and Page 6 line number 209-214).
In this study, we excluded participants with extremely low or high energy intake (<600kcal/day or >4000kcal/day). They were outliers and deviated from the energy intake population of the elderly that we were targeting. This reference value was chosen for the following reasons: <600 kcal/day is equivalent to half the calorie intake required for the lowest physical activity while >4000 kcal/day is equivalent to 1.5 times the energy intake required for medium physical activity (ref 43, Suzuki et al.).
Q3. The time of testing the blood pressure should be mentioned. For example, in the morning or at night?
Author response: Thank you for pointing this out. Blood pressure measurement was performed between 10:00 and 13:00. We have now included this information in the revised manuscript (Page 4 line number 134-136).
Q4. What’s the difference of the health condition between normal blood pressure and the participants who with history of hypertension? Will the health condition of the participants who with history was similar to the normal person after they recovery from hypertension? What’s the meaning of collecting the group of HP history group?
Author response: Thank you for raising this important point. To increase clarity, in the revised manuscript, we renamed the groups to normotensive, hypertensive, and history of hypertension (Page 4 line number 144-147). From the studies conducted previously (ref 5, Umemura et al), it can be inferred that the resistance of vessels is fundamentally different between the normotensive individuals and those with a history of hypertension. This might be related to the differences in lifestyles; in this study, lifestyle included not only dietary habits, but also all the other assessed factors. Therefore, even if the blood pressure of history of hypertension participants is maintained at normal level by medication, they cannot be considered in the same condition as the normotensive participants.
Q5. Please put the usage of medicine information from the participants, because the different kinds of medicine might affect the blood pressure.
Author response: Thank you for your suggestion; we have now included the information on the medications used by the study participants in the discussion (Page 12 line number 381-388) and appendix B.
Result
Q1. Please add the frequency of the exercise from the participants, due to the number of times and periods of exercise that might affect the blood pressure.
Author response: Thank you for your suggestion; we have added this information in the revised manuscript (Page 7 line number 264). From the data presented in Table 1 of the revised manuscript, it can be inferred that frequency and periods of exercise by the participants had no significant impact on hypertension.
Q2. The oral health data including the remaining teeth and posterior occlusion condition, but the lack of oral examination (ex. periodontal status, DMFT….) need to be mentioned. Suggesting recheck the definition of “oral health”, or change to other words.
Author response: Thank you for pointing out the unclarified issues in our manuscript. In this study, we conducted oral examinations as described in the materials and methods section (Page 5 line number 171-177).
Q3. Please explain the meaning of “Constant” in Table 5, and its p-value and odds ratio
Author response: “Constant” refers to the ”intercept”. We have revised it for clarity and added explanations for the p-values and odds ratios in Table 5 in the revised manuscript (Page 10-11 line number 329-337).
Q4. Please explain more about the relationship between oral health data and other influence factors.
Author response: Thank you for the suggestion. We have now included more explanation on the relationship between the oral health data and other influence factors in the revised manuscript (Page 10 line number 321-325).
Discussion
Q1. Due to one of the including criteria was BDHQ (energy>4000 kcal/day), but there was no upper limit or lower limit of the value, so is there any possible of the different hypertension or health condition with the participant who’s BDHQ was 5000 kcal/day, but another participant was 15000 kcal/day, which might influence the result? This issue should be discussed.
Author response: We thank for raising this important point. In this study, we excluded participants with extremely low or high energy intake (<600kcal/day or >4000kcal/day). This reference value was chosen based on criteria described in a previous study (ref 43, Suzuki et al.) where <600 kcal/day is equivalent to half the calorie intake required for the lowest physical activity while >4000 kcal/day is equivalent to 1.5 times the energy intake required for medium physical activity. Further, this criterion was set since in a study, it was demonstrated that extremely low or high calorie intake values may provide an improper response in the BDHQ survey. This information is now included in the revised manuscript (Page 3-line number 119-124).
Q2. The discussion of 4.2., the title was “The relationship of Oral health, Dietary Intake, and Hypertension”, but the part of oral health and the relationship with others were rarely discussed, please discuss more about the oral health and dietary intake (for example, oral moisture effects the diet intake?).
Author response: Thank you for the comment. As per your suggestion, the revised manuscript (Page 11-12 line number 365-400), we have included discussion on the relationships of oral health, dietary intake, and hypertension. We also elaborate on other factors that influenced hypertension (Page 12 line number 407-426).

Reviewer 2 Report
Thank you for the opportunity to review your work. Overall, the manuscript was interesting. However, several alterations are needed in order to improve the quality of the manuscript. Specific comments and suggestions are given below.
ABSTRACT
Line 21: In the ABSTRACT, the research background should be briefly described.
Line 22: Numbers should not be used at the beginning of a sentence.
Line 32: OR and 95% CI might be better than 1.7-folds.
INTRODUCTION
Overall, it would be helpful if the authors provided data on the prevalence of hypertension in older Japanese people, not in the general population. Also, the prevalence of impaired mastication is needed.
Line 51-66: The authors focus on the health effects of impaired mastication. There's no indication why the authors study on the association between dietary intake, impaired mastication, and hypertension and why they didn't consider other age-related chronic diseases such as diabetes or osteoporosis.
Line 79-87: The authors should improve this paragraph to clarify why this study is essential and innovative.
METHOD
Line 106-108: How about participants whose calorie intake is deficient?
Line 114: Why just rest for 1 min? Generally, it is better to rest for more than 5 min. Would this impact blood pressure?
Line 122-125: The names of the three groups are confusing. The authors can improve this part, describe the grouping method in detail, and it would be helpful to change the name of these three groups.
Line 181-190: Is BDHQ a semi-quantitative FFQ? How did the authors calculate the daily intake of nutrients and food groups (as you showed in Table 3)? Have reliability and validity tests been performed on sodium, potassium, and other nutrients from this BDHQ? It should be clear in the method.
RESULTS
Means, standard deviations, and the P-value should be reported to the same decimal digits throughout.
Table 2: It would be better to add a footnote of ‘kgf’ as ‘kilogram-force’ under the table.
Table 4: The nutrients intake and food group intake were associated with age, but not oral health. This is not consistent with the authors’ conclusion.
Line 280 – 296: In section 3.5, the authors discussed factors contributing to hypertension, which is also inconsistent with the conclusion (line 33 - 34) and the suggested hypotheses (line 86-87). The nutritional status of the study population is also unclear.
I suggest that authors use nutritional intake as dependent variables, increased blood pressure, reduced masticatory function as associated factors, and age, body mass index, and other factors as covariates.
DISCUSSION:
Line 318-319: It would be better to use ‘consider’ rather than ‘hypothesize.’
Line 320-335: In 4.1, the authors discussed the relationship between oral health on hypertension. But the age was significantly different within blood pressure groups. Is age the main reason of impaired mastication and hypertension?
Additionally, in lines 330-335, the explanation regarding ‘exceedingly high salt intake’ is inconsistent with the results, either with sodium or salt intake (Table 3). Why do the authors think individuals without posterior occlusion may consume excessive salt?
Line 371-375: Maybe the authors could discuss why this study failed to find the relationship between education and living arrangements with the hypertensive risk.
Moreover, the authors focus on the effect of oral health and dietary intake on hypertension, which is inconsistent with the suggested conclusion. Why was the expected outcome not achieved? Need more comparative discussion.
I also suggest the authors improve their DISCUSSION from these aspects: What Japanese government has done to improve oral health for the elderly? Challenges?
CONCLUSIONS:
LINE 387-390: This section should be enlarged and improved, reporting the main results and the future perspective of this study.
THANK YOU
Author Response
RE: (Nutrients-1604969)
Point-by-point responses to the Reviewer 2 comments
Abstract
Q1. Line 21: In the ABSTRACT, the research background should be briefly described.
Author response: We thank you for the suggestion. In the abstract section of the revised manuscript, we have included a brief background related to the topic of the research conducted in the present study.
Page 1 line number 21-23, in the revised manuscript, “Hypertension is related to impaired mastication that causes malnutrition, declining the general health of older adults. This study assessed the role of dietary intake in the relationship between oral health and blood pressure”.
Q2: Line 22: Numbers should not be used at the beginning of a sentence.
Author response: We thank you for noticing this. We have corrected this in the revised manuscript (Page 1 line number 23, in the revised manuscript).
Q3. Line 32: OR and 95% CI might be better than 1.7-folds.
Author response: Thank you for noticing this. In the revised manuscript, we have provided the actual value of OR (Page 1 line number 34). We could not add the 95% CI value in the abstract section due to word limitation; however, the data are available in the results section (Page 11 line number 332-336).
Introduction
Q4. Overall, it would be helpful if the authors provided data on the prevalence of hypertension in older Japanese people, not in the general population. Also, the prevalence of impaired mastication is needed.
Author response: As per your suggestion, we have now included data on the prevalence of hypertension in Japanese population (Page 2 line number 47-48) and impaired mastication (Page 2 line number 68-70) in the revised manuscript.
Q5. Line 51-66: The authors focus on the health effects of impaired mastication. There's no indication why the authors study the association between dietary intake, impaired mastication, and hypertension and why they didn't consider other age-related chronic diseases such as diabetes or osteoporosis.
Author response: We thank you for the comment. It is imperative that in old age patients, several diseases are linked to dietary intake which can also be affected by impaired mastication. However, for better presentation, in this manuscript, we focused on hypertension. Further, as we mentioned in the introduction section of the original submitted manuscript that the hypertension has become one of the leading causes of death in Japan, and the oral health is associated with hypertension. Hence, we performed this study to better understand the missing link of oral health, high blood pressure, and nutrition. In the revised manuscript, we have provided further justification on the aim of this study (Page 1 line number 43-48 and page 2 line number 52-65, 95-98).
Q6. Line 79-87: The authors should improve this paragraph to clarify why this study is essential and innovative.
Author response: We thank you for the comment and suggestion. In the revised manuscript (Page 2 line number 95-100), we have introduced text to highlight the significance and novelty of this study.
Method
Q7. Line 106-108: How about participants whose calorie intake is deficient?
Author response: In this study, we excluded participants with extremely low or high energy intake (<600kcal/day or >4000kcal/day). This reference value was chosen based on previous guidelines (ref 43, Suzuki et al.) that suggest that <600 kcal/day is equivalent to half the calorie intake required for the lowest physical activity while >4000 kcal/day is equivalent to 1.5 times the energy intake required for medium physical activity. This information is now included in the revised manuscript (Page 3 line number 119-124).
Q8. Line 114: Why just rest for 1 min? Generally, it is better to rest for more than 5 min. Would this impact blood pressure?
Author response: We thank you for noticing this mistake. In this study, the participants were allowed to rest for 10 min before the blood pressure measurements. The text is corrected in the revised manuscript (Page 4 line number 134-136).
Q9. Line 122-125: The names of the three groups are confusing. The authors can improve this part, describe the grouping method in detail, and it would be helpful to change the name of these three groups.
Author response: Thank you for the comment. In the revised manuscript (Page 4 line number 144-147), we have renamed (and defined) groups as “Normotensive” (SBP of <140 mm Hg and DBP <90 mm Hg), “Hypertensive” (SBP of ≥140 mm Hg and/or DBP ≥90 mm Hg), and “History of hypertension” (history of hypertension in the medical interview and/or is taking hypertension medication).
Q10. Line 181-190: Is BDHQ a semi-quantitative FFQ? How did the authors calculate the daily intake of nutrients and food groups (as you showed in Table 3)? Have reliability and validity tests been performed on sodium, potassium, and other nutrients from this BDHQ? It should be clear in the method
Author response: Thank you for pointing out the unclear issue. BDHQ is not a semi-quantitative FFQ. The calculations were made using fixed, sex-specific portion sizes of 58 food and beverage items and calculated using an ad hoc computer algorithm for the BDHQ. It has been validated for assessing energy intake and most nutrients intake in the adult Japanese population (ref 34 and 36, Kobayashi et al.). Further, in this study, we evaluated sodium to potassium ratio because studies have reported that high potassium intake or low sodium-to-potassium ratio may have beneficial possibilities on blood pressure. We have revised “Nutrition Assessment Methods” section in the revised manuscript (Page 6 line number 209-220) for clarity.
Result
Q10. Means, standard deviations, and the P-value should be reported to the same decimal digits throughout.
Author response: Thank you for the comment. We have revised the means, standard deviations, and the P-value to the same decimal digits in the revised manuscript.
Q11. Table 2: It would be better to add a footnote of ‘kgf’ as ‘kilogram-force’ under the table.
Author response: Thank you for the suggestion. Amended.
Q12.Table 4: The nutrients intake and food group intake were associated with age, but not oral health. This is not consistent with the authors’ conclusion.
Author response: Thank you for the comment. To address the concern, we have introduced explanation (Page 10 line number 321-325) in the revised manuscript.
“Although significant differences were found among the three groups, as seen in Table 3, age was the only factor significantly correlated with β-carotene, vitamin K, green and yellow vegetables, and meat. The sodium-to-potassium ratio was found to show a statistically significant correlation between the remaining teeth and masticatory performance; however, the correlation was weak”
Q13. Line 280 – 296: In section 3.5, the authors discussed factors contributing to hypertension, which is also inconsistent with the conclusion (line 33 - 34) and the suggested hypotheses (line 86-87). The nutritional status of the study population is also unclear.
I suggest that authors use nutritional intake as dependent variables, increased blood pressure, reduced masticatory function as associated factors, and age, body mass index, and other factors as covariates.
Author response: In this study, we assessed the role of oral health in nutrition and hypertension by exploring nutrient intakes of a Japanese elderly population. Hence, we do not think that it is necessary to change the objective and explanatory variables of the logistic regression analysis. However, to further increase the clarity on our objectives and the outcomes, we revised explanations in the Table 5 (Page 11 line number 338-341).
DISCUSSION:
Q14.Line 318-319: It would be better to use ‘consider’ rather than hypothesize.’
Author response: Amended.
Q15.Line 320-335: In 4.1, the authors discussed the relationship between oral health on hypertension. But the age was significantly different within blood pressure groups. Is age the main reason of impaired mastication and hypertension?
Author response: Age was both an explanatory variable and an adjustment variable in the logistic regression analysis, and the results we obtained are shown in the text (Page 11 line number 332-336). Impaired mastication and hypertension may also be affected by age, but we believe that it is one of the reasons, not the main reason. For clarity on this issue, we added explanations in the discussion section of the revised manuscript (Page 11 line number 365-377).
Q16. Additionally, in lines 330-335, the explanation regarding exceedingly high salt intake’ is inconsistent with the results, either with sodium or salt intake (Table 3). Why do the authors think individuals without posterior occlusion may consume excessive salt?
Author response: Thank you for highlighting this. The explanation was regarding the salt (sodium) intake. We considered that the individuals with absence of posterior occlusal contact will have difficulties in mastication, and hence, they would prefer easy-to-chew foods such as processed foods, which contain high sodium content. We have revised the introduction (Page 2 line number 66-78) and discussion (Page 12 line number 395-399, 411-417) sections to address the point in the revised manuscript.
Q17. Line 371-375: Maybe the authors could discuss why this study failed to find the relationship between education and living arrangements with the hypertensive risk.
Author response: We thank you for your suggestion. In the limitation section of the revised manuscript (Page 13 line number 448-450), we have included probable reasons on this aspect of the study.
“This might happen because we evaluated them based on their educational years rather than the educational level”
Q18. I also suggest the authors improve their DISCUSSION from these aspects: What Japanese government has done to improve oral health for the elderly? Challenges?
Author response: We again thank you for the constructive suggestion. We have now significantly revised the discussion section and also included the points indicated by you in the revised manuscript (Page 12 line number 375).
“Hence, the Japan Ministry of Health, Labour, and Welfare continuously promotes preserving ≥20 natural teeth until the age of 80 ” and
(Page 13 line number 427-435).
“Hence, the Japanese government conducted mass media-mediated public education, obligated food manufacturers to indicate salt content in food packaging, promoted nutritional labeling in school lunch/food vice industries, distributed home blood pressure measurements, and required all allied health professionals to instruct patients including non-hypertensive individuals to improve their lifestyle including the balanced dietary intake, improved oral hygiene, increased physical activities (approximately 1500 step increase in the number of steps), and maintain moderate alcohol consumption. Furthermore, the Japanese government provided home-visit dental services to promote the oral health of dependent older adults and covered dental care as part of its universal health coverage”
CONCLUSIONS:
Q19. LINE 387-390: This section should be enlarged and improved, reporting the main results and the future perspective of this study.
Author response: We have further revised the conclusion section (Page 13 line number 466-471) in the revised manuscript as per your suggestion. Thank you.

Round 2
Reviewer 1 Report
The author already revised the manuscript and clarified the question in detail, well done.
Author Response
The author already revised the manuscript and clarified the question in detail, well done.
Author response: We would like to thank you for your insightful comments on this paper. Your comments allowed us to significantly improve the paper.

Reviewer 2 Report
The authors have addressed most points raised by the reviewers, and the manuscript's quality has improved. However, there are a few issues that still need to be addressed.
Introduction:
Line 58-59: Although the authors explained clearly in the cover letter, they still did not clarify why they focused on hypertension and oral health in this version of the manuscript.
Line 114-122: These two sentences are contradicted. What exactly did the authors want to study? The role of nutrition or the role of oral health?
Methods:
Line 202: The first sentence is not suitable here.
Line 242: Dietary nutrition is one of the major variables in this study. Although there were several papers regarding BDHQ, it would be better to introduce this BDHQ in detail, at least the types of foods in BDHQ and the response options of the BDHQ.
Line 261-262: Why sample size was determined by the sodium intake furtherly?
Results:
Section 3.5 This title may not be suitable.
Table 5: I agree with the authors' keeping the logistic regression results. Furthermore, I suggest the authors add the analysis on the mediating effect of oral health between nutrition and hypertension since this is one of the most important results supporting the author's research hypothesis.
Discussion:
Line 413-430: The authors still did not answer the question of the role of oral health in nutrition and hypertension.
THANK YOU
Author Response
RE: (Nutrients-1604969)
Point-by-point responses to the Reviewer 2 comments
Introduction:
Line 58-59: Although the authors explained clearly in the cover letter, they still did not clarify why they focused on hypertension and oral health in this version of the manuscript.
Author response: Thank you for pointing out this issue to improve our manuscript. We have clarified why we focused on hypertension and oral health in the revised manuscript (Page 2, line 48-53).
Line 114-122: These two sentences are contradicted. What exactly did the authors want to study? The role of nutrition or the role of oral health?
Author response: Thank you for your logical question. The authors of the present study wanted to clarify the role of oral health in nutrition and hypertension since the study conducted by Fushida et al. did not consider the effect of masticatory performance on nutrition and just clarified the link between oral health and hypertension. To address this concern, we have revised the introduction to explain why we aimed to explore the role of oral health in nutrition and hypertension (Page 3 Line 101-109).
Methods:
Line 202: The first sentence is not suitable here.
Author response: Thank you for your suggestion, this sentence was deleted, and the manuscript was revised accordingly. (Page 5, Line 180-181)
Line 242: Dietary nutrition is one of the major variables in this study. Although there were several papers regarding BDHQ, it would be better to introduce this BDHQ in detail, at least the types of foods in BDHQ and the response options of the BDHQ.
Author response: Thank you for your suggestion. We have added the types of foods, and the response options of the participants included in the BDHQ, as seen in the “Nutrition Assessment Methods” section of the revised manuscript (Page 6, line number 220-233).
Line 261-262: Why sample size was determined by the sodium intake furtherly?
Author response: We would like to clarify that since sodium intake was found to be the factor that was closely related to BP of older adults in the previous study (ref 5, Umemura et al.), we found it appropriate to determine the sample size of this study based on sodium intake. (ref 36, Tamaki et al. and ref 22, Iwasaki et al.). To clarify this issue, we added explanations on the “Sample Size Calculation and Statistical analysis” section of the revised manuscript. (Page 7 line number 244-247)
Results:
Section 3.5 This title may not be suitable.
Author response: Thank you for your comment. In the revised manuscript, We have renamed it into “Impact factors affecting Hypertension”(Page 11 line number 350 and 359)
Table 5: I agree with the authors' keeping the logistic regression results. Furthermore, I suggest the authors add the analysis on the mediating effect of oral health between nutrition and hypertension since this is one of the most important results supporting the author's research hypothesis.
Author response: We think that you are recommending the use of mediation analysis. But a logistic regression analysis is the gold standard method for multivariate analysis in large cross-sectional surveys. Furthermore, since this is a cross-sectional study, there are many mediating variables other than nutrition, and it is necessary to construct a model combining moderated mediation. Hence, the authors think that it would be too complicated and inappropriate to represent the results of this study. However, we would like to consider this in the future, as mediation analysis may be possible by applying a multi-level mediation analysis or other innovations. Since our present study identified factors related to hypertension, it would be best to consider adding your suggestion to further clarify the role of oral health in nutrition and hypertension in our future longitudinal study.
Discussion:
Line 413-430: The authors still did not answer the question of the role of oral health in nutrition and hypertension.
Author response: Thank you for pointing out the unclear issue. We have included in the discussion the role of oral health in nutrition and hypertension of the study (Page 11-12 line number 382-391).